# GLI1 Deficiency Impairs the Tendon–Bone Healing after Anterior Cruciate Ligament Reconstruction: In Vivo Study Using *Gli1*-Transgenic Mice

**DOI:** 10.3390/jcm12030999

**Published:** 2023-01-28

**Authors:** Yake Liu, Shaohua Liu, Zhe Song, Daoyun Chen, Zoe Album, Samuel Green, Xianghua Deng, Scott A. Rodeo

**Affiliations:** 1Laboratory for Joint Tissue Repair and Regeneration, Orthopaedic Soft Tissue Research Program, Hospital for Special Surgery, New York, NY 10021, USA; 2Department of Orthopedic, Affiliated Hospital of Nantong University, Nantong 226001, China

**Keywords:** tendon–bone healing, hedgehog, *Gli1*, anterior cruciate ligament reconstruction

## Abstract

Hedgehog (Hh) signaling plays a fundamental role in the enthesis formation process and GLI-Kruppel family member GLI1 (*Gli1*) is a key downstream mediator. However, the role of *Gli1* in tendon–bone healing after anterior cruciate ligament reconstruction (ACLR) is unknown. To evaluate the tendon–bone healing after ACLR in *Gli1^LacZ/LacZ^* (GLI1-NULL) mice, and compare *Gli1^LacZ/WT^* (GLI1-HET) and *Gli1^WT/WT^* wild type (WT) mice, a total of 45 mice, 15 mice each of GLI1-NULL, GLI1-HET and WT were used in this study. All mice underwent microsurgical ACLR at 12 weeks of age. Mice were euthanized at 4 weeks after surgery and were used for biomechanical testing, histological evaluation, and micro-CT analysis. The GLI1-NULL group had significantly lower biomechanical failure force, poorer histological healing, and lower BV/TV when compared with the WT and GLI1-HET groups. These significant differences were only observed at the femoral tunnel. Immunohistology staining showed positive expression of Indian hedgehog (IHH) and Patched 1(PTCH1) in all three groups, which indicated the activation of the Hh signal pathway. The GLI1 was negative in the GLI1-NULL group, validating the absence of GLI1 protein in these mice. These results proved that activation of the Hh signaling pathway occurs during ACL graft healing, and the function of *Gli1* was necessary for tendon–bone healing. Healing in the femoral tunnel is more obviously impaired by *Gli1* deficiency. Our findings provide further insight into the molecular mechanism of tendon–bone healing and suggest that *Gli1* might represent a novel therapeutic target to improve tendon–bone healing after ACLR.

## 1. Introduction

Anterior cruciate ligament reconstruction (ACLR) has been widely performed to treat anterior cruciate ligament (ACL) injury [1]. Secure healing of the tendon graft in the bone tunnel is required for successful ACLR. Although the clinical outcomes of ACLR are generally good, delayed or incomplete healing between the tendon graft and bone can result in recurrent laxity. Furthermore, resorption of bone along the tunnel can lead to the clinical phenomenon of “tunnel widening” which may compromise graft fixation strength, potentially contributing to recurrent knee laxity and also complicate revision surgery [2]. The slow process of graft-to-bone healing contributes to the prolonged rehabilitation period, increased risk of re-injury and risk of long-term failure after ACLR [3,4]. The native tendon–bone enthesis consists of four layers that form a gradual fibrocartilaginous transition from tendon to bone, which is well adapted to transfer force [5]. After ACLR, the microstructure and composition of the native enthesis is not reformed. Rather, healing occurs by formation of a fibrovascular scar tissue interface, which has inferior material properties and may contribute to recurrent knee laxity. Therefore, further understanding of the mechanism of the tendon–bone healing process is necessary for the improvement of clinical outcomes following ACLR.

Among the signaling pathways that may play a role in the mechanism of tendon–bone healing, the hedgehog (Hh) signaling pathway is of special interest. The Hh family of proteins, such as Indian hedgehog (IHH), has been studied since the 1980s [6]. Recently, the Hh signaling pathway was reported to be active in the differentiating insertion site during tendon development [7,8]. Transgenic mice with loss of function of Hh signaling demonstrate significant histological, gene expression and biomechanical changes at tendon insertion sites when compared with normal mice [9]. Specifically, Breidenbach et al. confirmed that ablating Hh signaling reduced mineralized fibrocartilage in the enthesis of the normal tendon, resulting in decreased biomechanical strength [10].

Furthermore, some authors reported that Hh signaling was active at the tendon–bone interface during the healing process following ACLR or rotator cuff repair. Carbone et al. found that the Hh signaling pathway is active during ACLR healing and that pre-tensioning of the graft at the time of surgery resulted in increased Hh signaling expression [11]. Zong et al. reported that Hh signaling appeared to play an important role in the stromal cell-induced stimulation of fibrocartilage formation in a model of rotator cuff tendon healing [12]. Recently, Wada et al. reported that excessive tendon loading may delay the tendon healing process by affecting the activity of the Hh pathway in a murine rotator cuff repair model [13]. These findings led us to hypothesize that the Hh signal pathway might be fundamentally involved in the tendon–bone healing process. However, no previous studies directly investigated if a lack of the Hh signal pathway would have a significant impact on the tendon–bone healing process after ACLR.

Transgenic animals provide a powerful tool to evaluate the involvement of specific signaling pathways. The downstream gene of the Hh signal pathway, GLI-Kruppel family member GLI1 (*Gli1*) is a key effector in this pathway, which is upregulated and transported into the nucleus after activation of the Hh signaling cascade [14]. A clinially relevant murine ACLR model has been established which enables us to use genetically modified mice to study the biological mechanism of tendon–bone healing [15]. Therefore, the purpose of the present study is to evaluate tendon–bone healing after ACLR in *Gli1^LacZ/LacZ^* (GLI1-NULL) mice and compare them with *Gli1^LacZ/WT^* (GLI1-HET) and *Gli1^WT/WT^* wild type (WT) mice in order to gain further insight into the mechanism of the tendon–bone healing process.

## 2. Materials and Methods

### 2.1. Animals

All study procedures were approved by the Institutional Animal Care and Use Committee (IACUC #2018-0028). A commercially available line of *Gli1^tm2Alj^*/*J* mice (herein referred to as *Gli1^LacZ^* reporter mice) was purchased (The Jackson Laboratory, Bar Harbor, ME, USA), and the colony was bred and maintained in our institute for experimental purposes. The *Gli1^LacZ^* reporter mouse strain harbors a transgene which contains a genetic insertion of the *LacZ* gene, encoding beta-galactosidase into the first coding exon of the *Gli1* gene and replacing the genomic fragment encoding the entire N-terminal and zinc-finger domains of the targeted locus (exons 2–7), thereby abolishing endogenous *Gli1* gene function even if alternative splicing occurs. To expand the mouse colony, *Gli1^LacZ/WT^* heterozygotes were crossed with C57BL/6 wild-type mice to create a breeding population, from which *Gli1^LacZ^* homozygotes (*Gli1^LacZ/LacZ^*: GLI1-NULL), heterozygotes (*Gli1^LacZ/WT^*: GLI1-HET) and wild type mice (*Gli1^WT/WT^*: WT) were generated. Theoretically, the *Gli1* gene will be functional in the GLI1-HET and WT mice and will be lost in the GLI1-NULL mice.

Mice underwent tail snipping and polymerase chain reaction (PCR) testing for genotype identification by using a genotyping protocol provided by the vendor (The Jackson Laboratory, Bar Harbor, ME, USA) and the following primer sequences: *Gli1* Common (oIMR7888), *Gli1* Mutant Reverse (oIMR8770) and *Gli1* Wild Type Reverse (oIMR9034).

### 2.2. Study Design

A total of 45 mice, 15 mice/genotype (WT, GLI1-HET, GLI1-NULL), were used in this study. At 12 weeks of age, all mice underwent unilateral microsurgical ACLR with flexor digitorum longus (FDL) tendon autografts as previously described [15,16]. Mice were euthanized at 4 weeks after surgery and the retrieved tissues were used for evaluation (Figure 1).

### 2.3. Biomechanical Testing

Seven specimens per genotype (WT, GLI1-HET, GLI1-NULL) were prepared for biomechanical testing using an Instron Materials Testing System (Norwood, MA, USA) as described previously [16]. Knee specimens were carefully dissected using surgical magnifying loupes. The femur and tibia were independently placed in 2.0-mL tubes of Bondo Lightweight Filler 265 (3 M), and all soft-tissue structures other than the graft were dissected. The surgical clip and suture used for the initial graft fixation were removed. The specimen was placed into the biomechanical testing device in a position that allowed distraction in line with the axis of the graft, to allow tensile testing to failure. The specimen was loaded to failure at a rate of 10 mm/min (0.167 mm/s). Failure force (N) and stiffness (N/mm) were measured from the load–displacement curve. The site of graft failure (femoral tunnel, midsubstance or tibial tunnel) was recorded.

### 2.4. Histologic Evaluation and Immunohistochemical (IHC) Staining

Four mice per genotype were used for histological evaluation. Tissue specimens were fixed in 10% neutral-buffered formalin for 48 h and then decalcified, dehydrated and embedded in paraffin. Serial cross-sections of the femur and tibia were obtained and stained with hematoxylin and eosin (H&E), safranin-O, and picrosirius red. Quantitative histologic evaluation of the tendon–bone interface was based on a previously reported scoring system [17,18]. All measurements were performed by two investigators, who were blinded to the groups. We examined 8–10 fields of view for each specimen.

Sections were also used for immunohistochemical staining using specific primary antibodies against GLI1 (NB600-600, purified rabbit polyclonal IgG, Novus Biologicals, Long Island City, NY, USA), IHH (ab39634, purified rabbit polyclonal IgG, Abcam Plc., Cambridge, MA, USA) and PTCH1 (ab53715, purified rabbit polyclonal IgG, Abcam Plc., Cambridge, MA, USA). A secondary antibody (ImmPRESS Anti-Rabbit Ig Kit, Vector Laboratories, Inc., Burlingame, CA, USA) and the Dako Liquid DAB+ Substrate Chromogen System (K3468, Agilent, Santa Clara, CA, USA) were used following the immunostaining procedures as recommended by the manufacturer, and the sections were counterstained with hematoxylin. Negative controls using just the secondary antibody and all subsequent reagents were used to verify the absence of non-specific staining. Digital images of the stained tissue sections were taken using a digital single-lens reflex camera (D3400, Nikon Inc., Melville, NY, USA) and analyzed with Image J software (Image J 1.51j8, National Institutes of Health, Bethesda, MD, USA) within a region of interest (ROI) set at the interface between graft and bone. Automated scoring of the IHC tissue sections was done using a previously described protocol. All digital images were analyzed by using IHC Profiler, an open-source plugin on ImageJ for the quantitative evaluation and automated scoring of immunohistochemistry images [19]. The intensity was assigned a score as high positive (3+), positive (2+), low positive (1+), and negative (0). The illumination and detection parameters of the microscope were kept constant among specimens to allow for direct comparisons.

### 2.5. Micro-CT Examination

A micro-CT was performed to evaluate the bone parameters using a high-resolution specimen micro-CT system (Scanco µCT 35, Scanco Medical, Brüttisellen, Switzerland). Micro-CT imaging was performed in the axial plane with a slice thickness of 6 µm at 55 kVp and 145 µA. The volume of bone within the tunnel was measured at the aperture, mid-tunnel and tunnel exit, respectively, and was reported as bone volume (BV), total volume (TV) and directly measured bone volume fraction (BV/TV [voxels/mm^3^]). Scanco mCT software (DECwindows Motif 1.6; Hewlett-Packard, Palo Alto, CA, USA) was used for viewing, image analysis and 3-dimensional reconstruction. A cylindrical volume of interest with a diameter of 0.64 mm (the same diameter as the needle used to create the tunnels) and a height of 30 slices’ thickness was analyzed for each level.

### 2.6. Statistical Analysis

All descriptive statistics were reported as means and standard deviations. GraphPad Prism 7 (GraphPad Software, La Jolla, CA, USA) was used for the statistical calculations. A one-way ANOVA with Tukey’s honest significant difference test for post hoc multiple comparisons or Kruskal–Wallis test was used to analyze the results of the histological quantification, micro-CT and the biomechanical test among the three groups. The significance level was set at *p* < 0.05 for each comparison.

## 3. Results

### 3.1. Biomechanical Evaluation

The failure force was 2.97 ± 0.79 N in the WT group, 3.14 ± 0.70 N in the GLI1-HET group and 1.91 ± 0.81 N in the GLI1-NULL group. There were significant differences among the three groups (*p* = 0.01). The failure forces of the WT group and GLI1-HET were significantly higher than those in the GLI1-NULL group (*p* = 0.04 and 0.02, respectively). (Figure 2A) There were no significant differences in stiffness among the groups. (WT: 3.14 ± 1.30; GLI1-HET: 3.34 ± 1.00; GLI1-NULL: 2.89 ± 1.70; *p* = 0.82) (Figure 2B) There were 17 among 21 (81%) samples that failed by pull-out from the femoral tunnel during tensile testing (Table 1).

### 3.2. Histological Evaluation

Evaluation of the healing tendon–bone interface demonstrated inferior continuity between the tendon graft and bone tunnel in the GLI1-NULL group compared with the GLI1-HET and WT groups at 4 weeks after ACLR. (Figure 3) In the GLI1-NULL group, there was obvious fibrovascular tissue in the tendon–bone connection on both the femoral and tibial sides, and a wider and more distinct interface was present compared with the other groups. In the WT group and GLI1-HET group, a smooth transition from bone to the tendon with some fibrocartilage-like cells was present.

Safranin-O staining showed some fibrocartilage-like cells at the tibial tunnel interface in the WT group and GLI1-HET group, while rare cartilage-like cells could be seen in the GLI1-NULL group at either femoral or tibial sides (Figure 4).

Picrosirius red staining showed that oblique or perpendicular collagen fibers that crossed the tendon–bone interface were formed between the tendon graft and bone tunnel in the WT group and GLI1-HET group. There were less organized and fewer connecting collagen fibers at the tendon–bone interface in the GLI1-NULL group (Figure 5).

Histologic scores of the tendon–bone interface for the three groups are presented in Figure 6. The WT group had a higher score than the other two groups. The WT group had a significantly higher score on the femoral side when compared with the GLI1-NULL group (*p* = 0.04). There was no significant difference among groups on the tibial side.

Immunohistological staining for IHH and PTCH1 was positive at the interface in all the groups (Figure 7 and Figure 8). However, GLI1 was positive at the interface only in the WT and GLI1-HET groups, and was negative in the GLI1-NULL group. This expression difference was significant on the femoral side (Figure 9). These results indicated that although the Hh signaling pathway was activated, as expected, due to knock-in of *LacZ* in the *Gli1* gene, its expression was lost in the GLI1-NULL group.

### 3.3. Micro-CT

Micro-CT examination showed bone tissue formation surrounding the graft within the tunnel (Figure 10). Less surrounding bone formation was observed in the GLI1-NULL group. There were significant differences among the WT, GLI1-HET and GLI1-NULL groups in BV/TV at the aperture of the femoral tunnel, and the WT group had a significantly higher value when compared with the GLI1-NULL group (*p* = 0.03).

## 4. Discussion

The primary finding of this study was that the tendon–bone healing process was impaired in mice lacking *Gli1,* the downstream gene and the key effector of the Hh signal pathway; this was found by using transgenic *Gli1^LacZ^* mice in a clinically relevant ACLR model. At 4 weeks after surgery, the GLI1-NULL group had inferior healing at the tendon–bone interface, less surrounding bone formation and lower biomechanical failure force when compared with the WT and GLI1-HET groups. Immunohistology staining showed positive expression of IHH and PTCH1 in all three groups, which indicated the activation of the Hh signal pathway. However, GLI1 expression was absent in the GLI1-NULL group. These results confirmed that the deficiency of *Gli1* had an adverse effect on tendon–bone healing after ACLR, even if the upstream Hh signaling pathway was activated. Of note, the significant histological and radiological differences were only detected on the femoral side.

Secure tendon–bone healing is critically important for a successful ACLR. The normal enthesis is a highly specialized tissue, with a gradual transition from tendon to nonmineralized fibrocartilage, mineralized cartilage and bone, that is well-adapted to transfer force [4,18]. However, healing of the tendon graft in a bone tunnel occurs via the formation of a fibrovascular scar tissue interface with inferior material properties which may contribute to recurrent knee laxity and later failure. Thus, there is a need to understand the basic mechanism(s) of insertion site formation in order to identify methods to improve the microstructure and composition of the healing attachment site following surgical reconstruction.

Insight into soft tissue attachment site healing can be gained from study of signaling molecules involved in enthesis formation during embryologic development. Hh signaling is known to play a fundamental role in enthesis formation. Hh proteins are secreted paracrine ligands which bind to the membrane-bound receptor Patched 1 (PTCH1), which subsequently causes disinhibition of the transmembrane protein and results in signaling-cascade activation and upregulation of the expression of *Gli1* [14]. *Gli1* is the key effector in this pathway; it is transported into the nucleus after Hh signal cascade activation and mediates further functional pathways [20]. Specifically, Hh signaling plays an important role in fibrocartilage formation in the developing enthesis [9,10]. Liu et al. found that transgenic mice with absent IHH signaling knockout have significant histological, gene expression and biomechanical changes at tendon insertion sites when compared with normal mice [9]. Another study also confirmed that ablating IHH signaling reduces mineralized fibrocartilage in the enthesis of the normal tendon, resulting in decreased biomechanical strength [10]. Recently, Hh signaling activity was shown to be active during the tendon–bone healing process [11,12,21]. In this study, we also found positive expressions of IHH, one of the Hh proteins, and PTCH1 at the healing tendon–bone interface in all three groups, which indicates the activation of Hh signaling after ACLR. However, the role of Hh signaling activity in tendon–bone healing after ACLR is unknown.

The histological, radiological and functional (biomechanical) outcomes of this study demonstrated that the GLI1-NULL group had inferior tendon–bone healing compared with WT and GLI1-HET groups. In the GLI1-NULL group, there was obvious fibrovascular tissue in the tendon–bone connection, and a wider and more distinct interface was observed when compared with the other groups with functioning GLI1. In the WT group, a smooth transition from bone to the tendon with some fibrocartilage-like cells were noticed, and oblique or perpendicular collagen fibers that crossed the tendon–bone interface were formed between the tendon graft and bone tunnel (Figure 3, Figure 4 and Figure 5). Furthermore, less surrounding bone formation and significantly lower failure force were observed in the GLI1-NULL group. These results indicate the fundamental role of Hh signaling in the tendon–bone healing process. Furthermore, our finding that GLI1 was not expressed in the GLI1-NULL group validated the absence of GLI1 protein in these mice and proved that even if the upstream Hh signaling pathway was activated after ACLR, worse tendon–bone healing was achieved in the setting of *Gli1* deficiency.

A notable finding in this study is that significant differences were only observed at the femoral tunnel. The GLI1-NULL group had a significantly lower histological score on the femoral side when compared with the WT group, and the difference in BV/TV between GLI1-NULL and WT groups was only significant at the aperture of the femoral tunnel. Some prior studies have evaluated the difference in healing between femoral and tibial tunnels. Berg et al. reported that bone healing was slower and incomplete in the articular segment of the tunnel closest to the joint surface, and the femoral side was inferior to the tibial side in the vivo rabbit model [22]. Similarly, Nakase et al. found that healing in the tendon–bone interface of the femoral tunnel was slower than that of the tibial tunnel based on magnetic resonance imaging (MRI) examination of human patients who underwent arthroscopic ACLR [23]. Studies in our laboratory using a rabbit ACLR model found that ingrowth of new bone around the tendon graft in the femoral tunnel was inversely proportional to the magnitude of graft tunnel motion, with slower ingrowth and a wider tendon–bone interface at the tunnel aperture [24]. Other studies also evaluated the effect of mechanical load on tendon–bone healing and suggested that there was progressively greater graft-tunnel motion as the distance from the fixation point increases, resulting in greater graft-tunnel motion at the tunnel aperture than at the tunnel exit for a graft fixed outside of the tunnel (suspensory fixation) [25,26,27,28]. These findings corresponded to our result that most samples failed at the femoral tunnel rather than the graft mid-substance or tibial tunnel, which implies that the femoral tunnel, especially the aperture site, might be more vulnerable to excessive graft-tunnel motion or mechanical load; this may lead to inferior tendon–bone healing.

Several prior studies have evaluated the relationship between mechanical load and the Hh signaling pathway. Carbone et al. found that the Hh signaling pathway is active and appears to be mechanosensitive during the tendon–bone healing process, as pre-tensioning of the graft at the time of surgery resulted in increased Hh signaling expression [11]. Wada et al. found that excessive postoperative tendon loading may delay the tendon–bone healing process by affecting the activity of the Hh pathway [13]. These results combined with the current study indicate that the Hh signaling via its downstream gene *Gli1* is affected by the local mechanical environment around the healing graft, which might also explain the difference in expression of GLI1 between the femoral and tibial tunnels. Given our finding that the GLI1-NULL group had significantly inferior tendon–bone healing in the femoral tunnel compared with the WT and GLI1-HET groups, *Gli1* could be a promising target to improve tendon–bone healing by ameliorating the detrimental effects of early aggressive mechanical load, thus allowing early rehabilitation. Further studies are needed to identify the mechanosensitivity of the Hh signaling pathway.

Although we found that the activation of the Hh signaling pathway and the function of *Gli1* play an important role following ACLR based on histologic, radiologic and biomechanical criteria, several limitations should still be noted. First, the postoperative observation period of 4 weeks was relatively short, and we only evaluated one-time point. More conclusive data will be provided by study of multiple and longer-term observations. Another limitation is the use of a small animal study using quadrupeds and with knee biomechanics that differ from those of human patients. Lastly, the underlying reasons for the significant differences in tendon–bone healing and GLI1 expression were only observed in the femoral tunnel, and still need to be explored and confirmed.

## 5. Conclusions

This study demonstrated that activation of the Hh signaling pathway occurs during ACL graft healing, and the function of *Gli1* was necessary for tendon–bone healing. Healing in the femoral tunnel is more obviously impaired by *Gli1* deficiency.

## Figures and Tables

**Figure 1 jcm-12-00999-f001:**
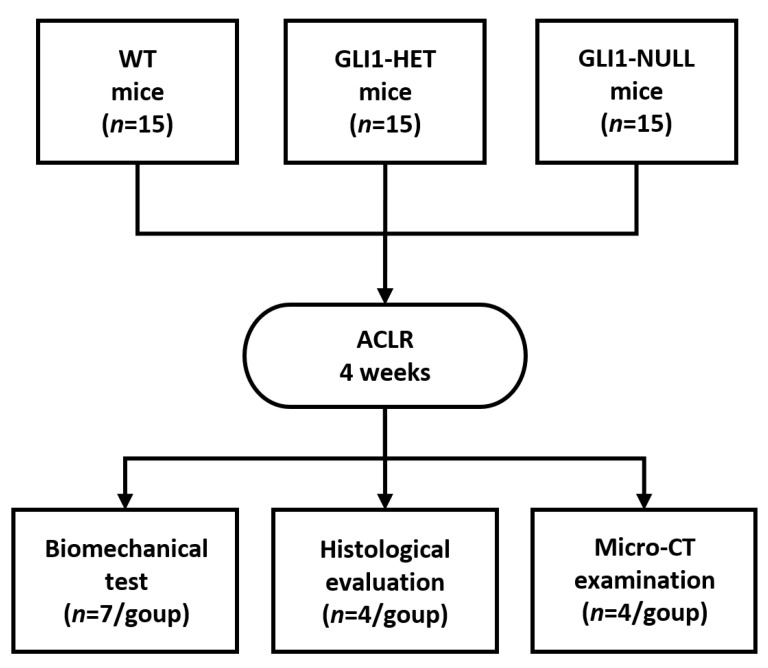
Study design.

**Figure 2 jcm-12-00999-f002:**
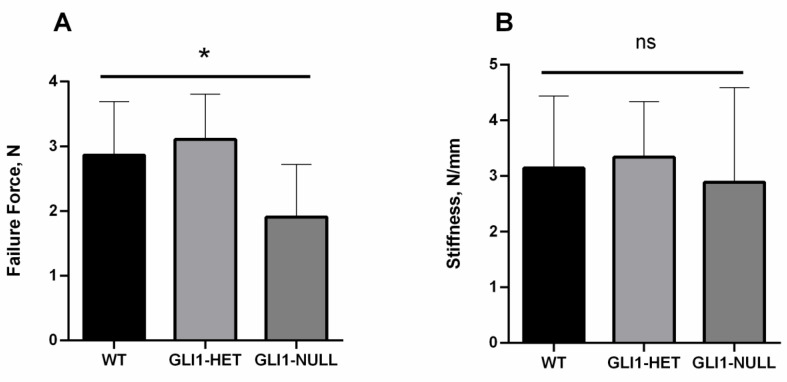
Biomechanical testing results of wild-type, GLI1-HET and GLI1-NULL mice at 4 weeks after ACLR. (**A**) The failure force in the GLI1-NULL group was significantly lower than the GLI1-HET or WT counterparts at 4 weeks after surgery. (**B**) Stiffness was comparable among groups. Values are presented as mean ± SD. *, *p* < 0.05.

**Figure 3 jcm-12-00999-f003:**
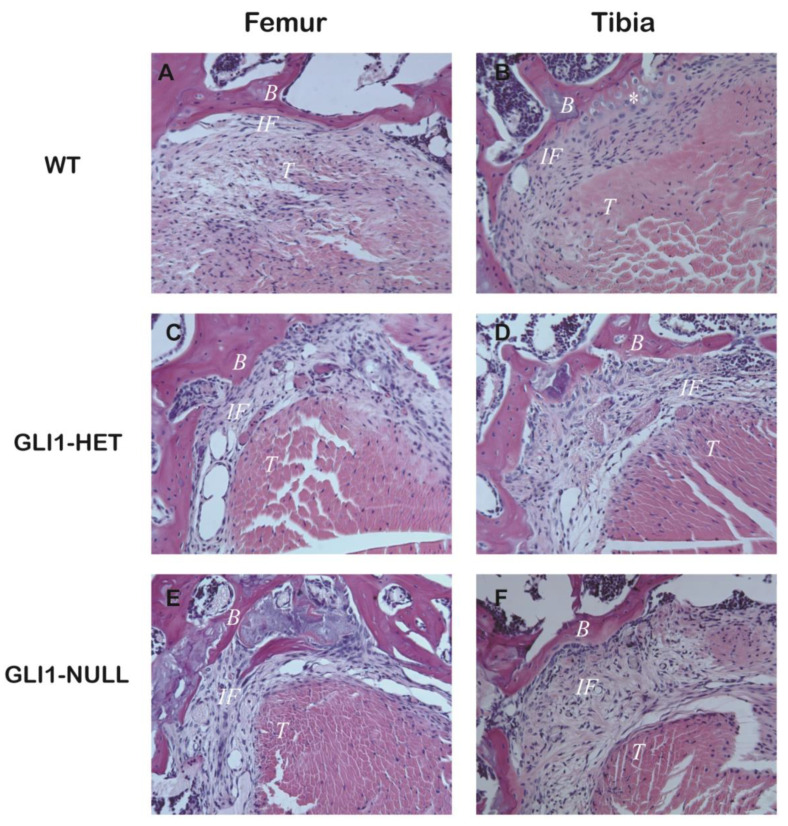
Hematoxylin and eosin staining of the tendon–bone interface of wild-type (**A**,**B**), GLI1-HET (**C**,**D**), and GLI1-NULL (**E**,**F**) mice at 4 weeks after ACLR (20×). Obvious fibrovascular tissue and wider and more distinct interface in the tendon–bone connection at both femoral and tibial sides could be observed in the GLI1-NULL group. (B: Bone tissue, IF: Interface, T: Tendon graft).

**Figure 4 jcm-12-00999-f004:**
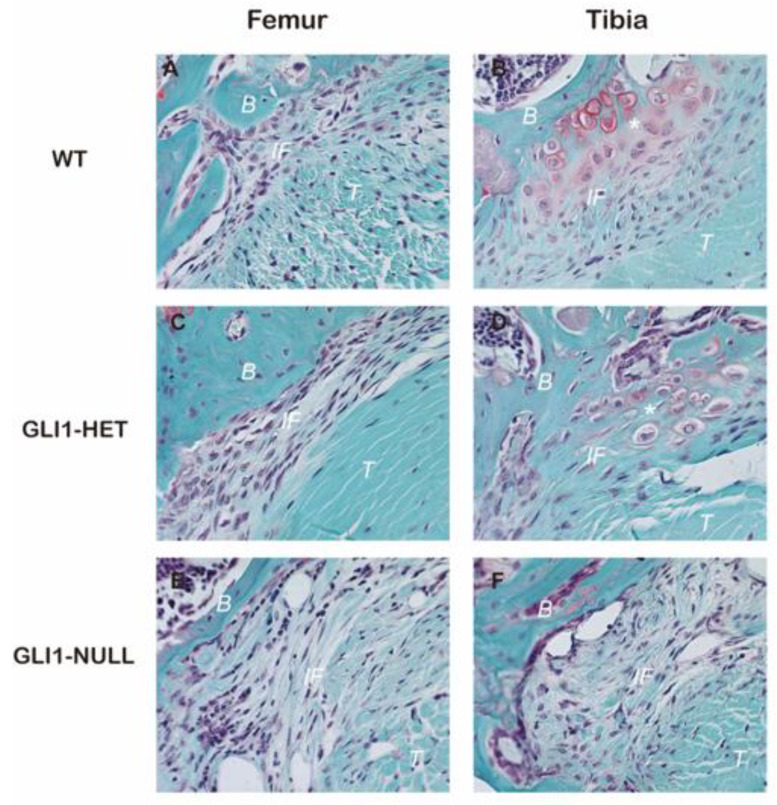
Safranin-O staining of the tendon–bone interface of wild-type (**A**,**B**), GLI1-HET (**C**,**D**) and GLI1-NULL (**E**,**F**) mice at 4 weeks after ACLR (40×). Fibrocartilage-like cells were observed at the tibial tunnel interface in the WT group and Gli1-HET group. (B: Bone tissue, IF: Interface, T: Tendon graft, *: Fibrocartilage-like cell).

**Figure 5 jcm-12-00999-f005:**
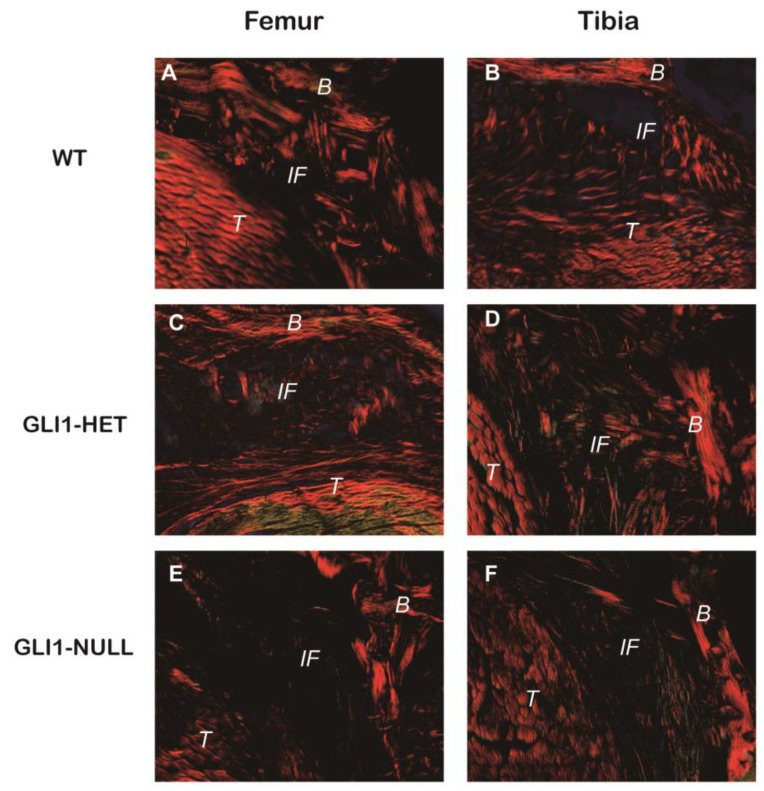
Picrosirius red staining of the tendon–bone interface of Wild-Type (**A**,**B**), GLI1-HET (**C**,**D**), and GLI1-NULL (**E**,**F**) mice at 4 weeks after ACLR (40×). Oblique or perpendicular collagen fibers that crossed the tendon–bone interface were formed in the WT group and GLI1-HET group, while there were minimal, poorly organized connecting collagen fibers at the interface in the GLI1-NULL group. (B: Bone tissue, IF: Interface, T: Tendon graft).

**Figure 6 jcm-12-00999-f006:**
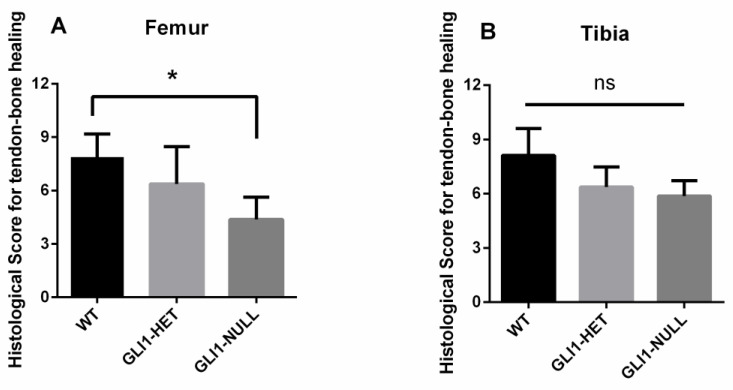
Histologic scores for tendon–bone healing at femoral (**A**) and tibial (**B**) sides in wild-type, GLI1-HET and GLI1-NULL mice at 4 weeks after ACLR. Values are presented as mean ± SD. The WT group had a significantly higher score on the femoral side when compared with the GLI1-NULL group (*p* = 0.04).

**Figure 7 jcm-12-00999-f007:**
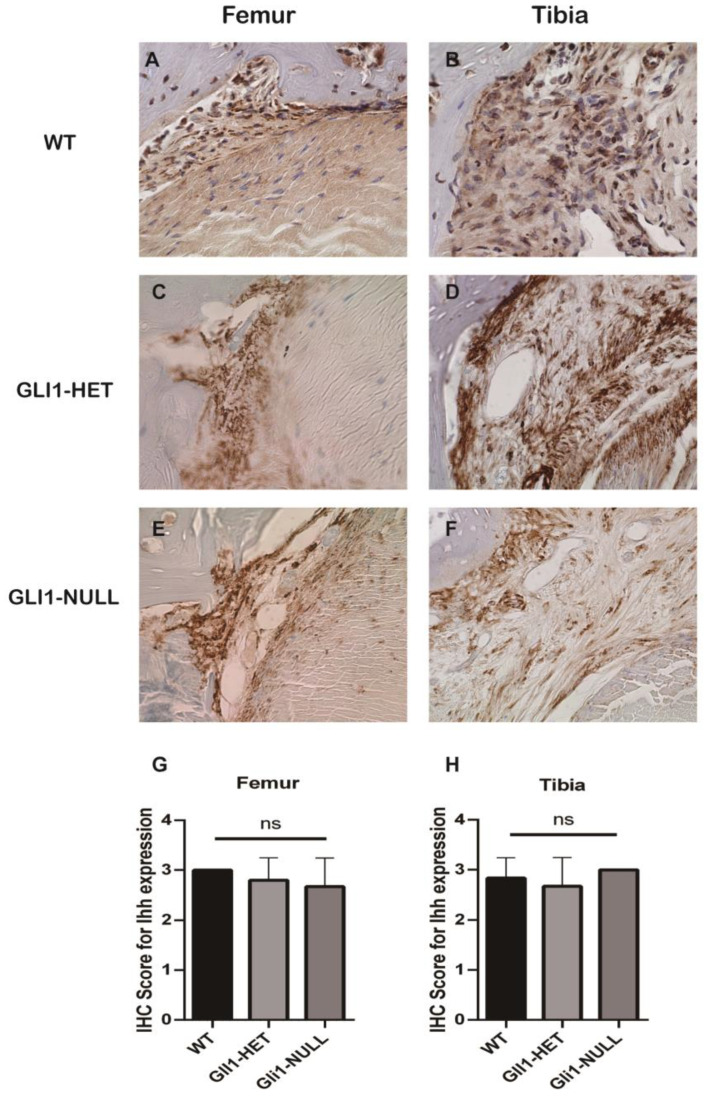
Representative images of IHH immunohistological staining of the tendon–bone interface of wild-type (**A**,**B**), GLI1-HET (**C**,**D**) and GLI1-NULL (**E**,**F**) mice at 4 weeks after ACLR (40×). All three groups had positive IHH expression, with no significant difference among groups (**G**,**H**). Values are presented as mean ± SD.

**Figure 8 jcm-12-00999-f008:**
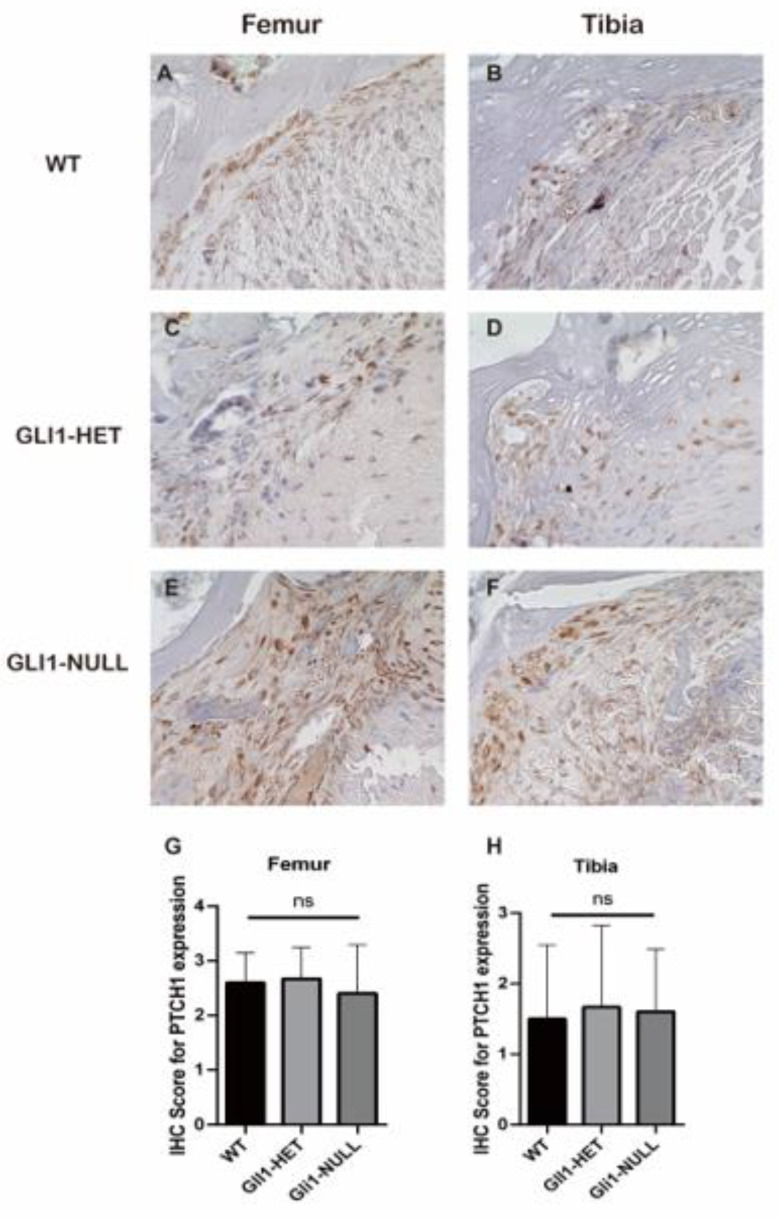
Representative images of PTCH1 immunohistological staining of the tendon–bone interface of wild-type (**A**,**B**), GLI1-HET (**C**,**D**) and GLI1-NULL (**E**,**F**) mice at 4 weeks after ACLR (40×). All three groups had positive PTCH1 expression, with no significant difference among groups (**G**,**H**). Values are presented as mean ± SD.

**Figure 9 jcm-12-00999-f009:**
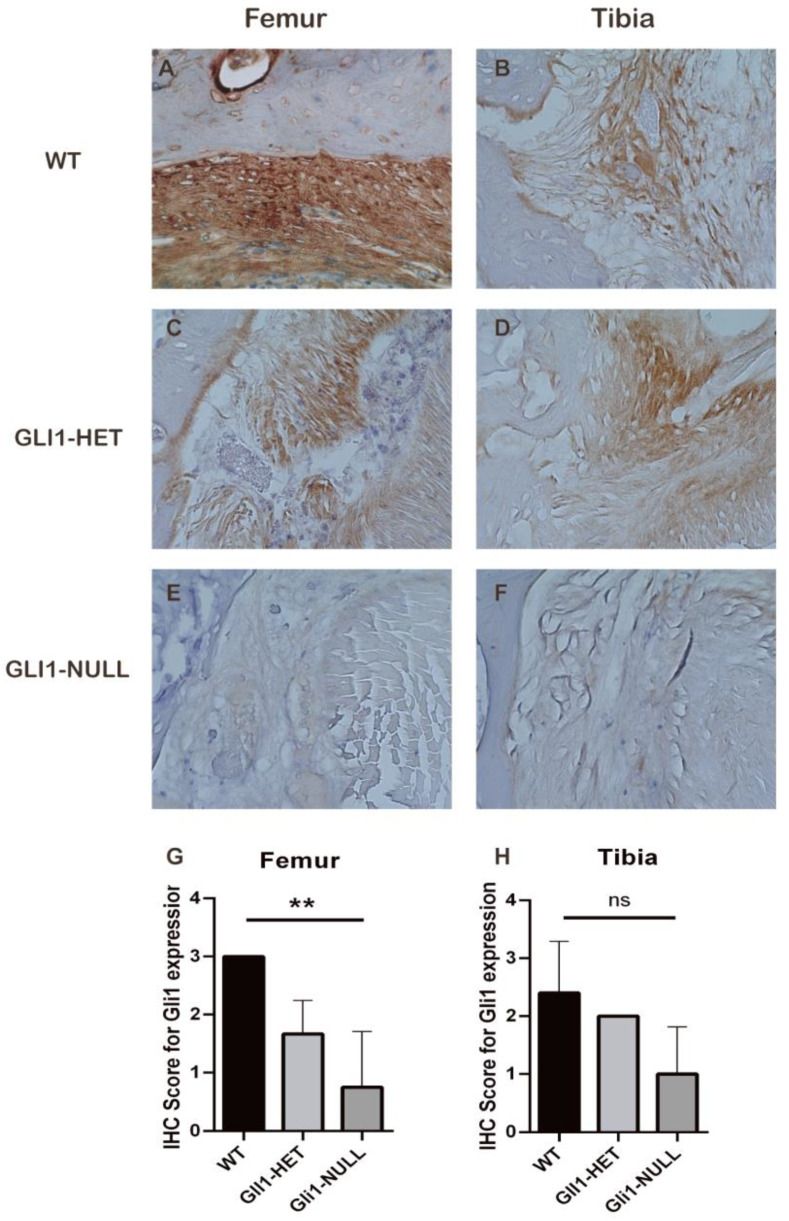
Representative images of GLI1 immunohistological staining of the tendon–bone interface of wild-type (**A**,**B**), GLI1-HET (**C**,**D**) and GLI1-NULL (**E**,**F**) mice at 4 weeks after ACLR (40×). GLI1 was positive at the interface only in the WT and GLI1-HET groups and was negative in the GLI1-NULL group. This difference was significant on the femoral side. (**G**,**H**) Values are presented as mean ± SD. **, *p* < 0.01.

**Figure 10 jcm-12-00999-f010:**
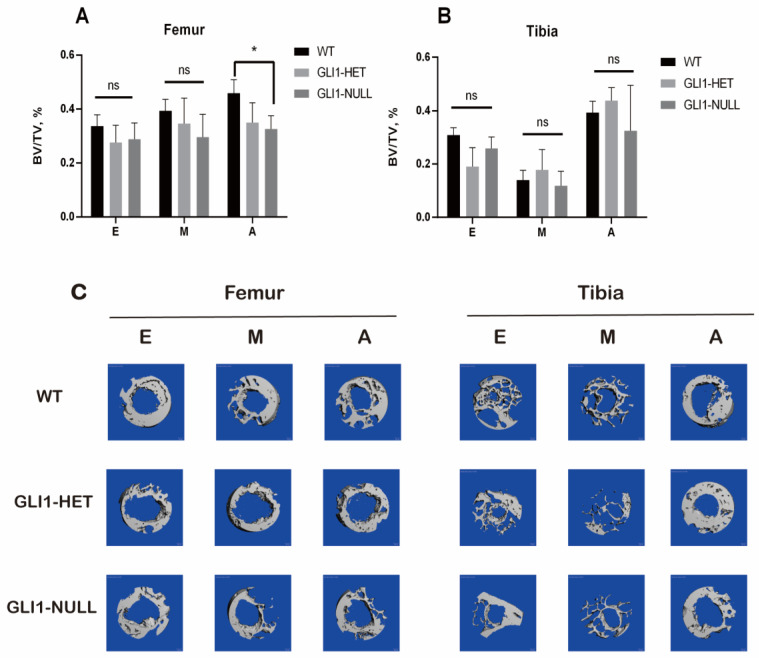
Results of micro-CT analyses for wild-type, GLI1-HET, and GLI1-NULL mice at 4 weeks after ACLR (40×). (**A**,**B**) Bone volume/total volume (BV/TV) in all three groups. Values are presented as mean ± SD. *, *p* < 0.05. (**C**) 3D-reconstruction images of all three groups (E: tunnel exit, M: mid-tunnel, A: aperture).

**Table 1 jcm-12-00999-t001:** Failure modes of all specimens (*N* = 7/group).

Failure Mode	WT	GLI1-HET	GLI1-NULL
Pull out from femur	5	6	6
Pull out from tibia	1	0	1
Midsubstance failure	1	1	0

## Data Availability

The datasets generated and analyzed during the current study are available from the corresponding author on reasonable request.

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
