# Peer review of "GLI1 Deficiency Impairs the Tendon–Bone Healing after Anterior Cruciate Ligament Reconstruction: In Vivo Study Using Gli1-Transgenic Mice"

_jcm, 2023, doi:10.3390/jcm12030999_

Round 1
Reviewer 1 Report
Liu et al. report in this paper that GLI-Kruppel family member GLI1 is a key factor for tendon-bone healing after anterior cruciate ligament reconstruction (ACLR), particularly in the femur tunnel. Tendinopathy is a serious health problem with many open questions. GLI1 is a key factor in the Hedgehog (Hh) signaling which is known to be important for the enthesis formation process of tendons. So, it is somehow not really surprising that GLI1-deficient mice have an impaired tendon-bone healing capacity. Yet, it is nevertheless important for the field to have a confirmative study for this pathway.
The experimental approach is well-chosen, the experiments are well-done and clearly presented in the figures.
In my opinion the following improvements should be added or included by the authors:
1) The data of the study do not fully support the claims/conclusions made by the authors. GLI1 deficient mice have obviously an impaired healing process but effect is by far not so dramatic/impressive to conclude that this factor is so essential for the process.
The authors provide no new molecular mechanism(s) for tendon-bone healing as stated in the abstract and discussion. At least in the discussion the reader what like to learn about the function of GLI1 in tendon repair.
Why and how could GLI1 be a therapeutic target for tendon healing?
2) The results section needs to be improved. In its current version it is so minimalistic that only a few sentences describe the data of 10 Figrues.
3) “rusults” in the Abstract, was very irritating. The paper needs a re-check and proof-reading.
Author Response
Liu et al. report in this paper that GLI-Kruppel family member GLI1 is a key factor for tendon-bone healing after anterior cruciate ligament reconstruction (ACLR), particularly in the femur tunnel. Tendinopathy is a serious health problem with many open questions. GLI1 is a key factor in the Hedgehog (Hh) signaling which is known to be important for the enthesis formation process of tendons. So, it is somehow not really surprising that GLI1-deficient mice have an impaired tendon-bone healing capacity. Yet, it is nevertheless important for the field to have a confirmative study for this pathway.
The experimental approach is well-chosen, the experiments are well-done and clearly presented in the figures.
In my opinion the following improvements should be added or included by the authors:
1) The data of the study do not fully support the claims/conclusions made by the authors. GLI1 deficient mice have obviously an impaired healing process but effect is by far not so dramatic/impressive to conclude that this factor is so essential for the process.
The authors provide no new molecular mechanism(s) for tendon-bone healing as stated in the abstract and discussion. At least in the discussion the reader what like to learn about the function of GLI1 in tendon repair.
Author’s response: In line 314-320 we briefly review the role of hedgehog signaling on development of the tendon-bone enthesis, which is relevant to this study where we are examining healing at the tendon-bone interface. Although we could discuss the role of Gli1 in tendon, that is somewhat beyond the scope and focus of the current study. If the editor believes that a review of hedgehog signaling in tendon is required, this can be added. Line 314-320 of our manuscript addresses the function of GLI1 in enthesis formation, as is copied below:
“Specifically, Hh signaling plays an important role in fibrocartilage formation in the developing enthesis 3, 11. Liu et al found that transgenic mice with absent IHH signaling knockout have significant histological, gene expression, and biomechanical changes at tendon insertion sites when compared to normal mice11. Another study also confirmed that ablating IHH signaling reduces mineralized fibrocartilage in the enthesis of the normal tendon, resulting in decreased biomechanical strength3. Recently, Hh signaling activity was shown to be active during the tendon-bone healing process5, 27, 28.”
Why and how could GLI1 be a therapeutic target for tendon healing?
Author’s response: The point of this statement is simply that identification of important signaling molecules will suggest avenues for further research. There are currently no known drugs or other pharmaceutical agents that we are aware of that affect hedgehog signaling. However, our data clearly show that GLI1 could be a therapeutic target for tendon-bone healing and is thus worthy of further research.
2) The results section needs to be improved. In its current version it is so minimalistic that only a few sentences describe the data of 10 Figrues.
Author’s response: It is not entirely clear to us how further description of the histologic results would improve the manuscript. We sequentially describe the results of hematoxylin and eosin staing (Figure 3), Safranin-O staining (Figure 4), picrosirius red staining (Figure 5), and then the quantitative score (Figure 6). Similarly, the immunohistochemistry data are described followed by both histologic images and graph of the quantitative scoring for each of Ihh, Patched 1, and Gli1. In general, we believe that histologic figures with an explanatory legend, as we have provided, is the optimal way to present histology results. The same can be said for the microCT data.
Perhaps the reviewer can specify what they think should be added to the Results section.
3) “rusults” in the Abstract, was very irritating. The paper needs a re-check and proof-reading.
Author’s response: We apologize for this single typographical error. The paper has been proof-read, and the necessary change made in the word “results” in the abstract (line 42).

Reviewer 2 Report
The manuscript presented for review is an interesting study on the molecular basis of recovery after ACLR. However, I have some questions and comments:
1. There is no information in the chapter ‘’Biomechanical testing’’ about the type of test used in the research. In the description of the results there is information that it is a tensil test. There is also no name of the equipment used.
2. How many slides and fields of view from each animal were evaluated for histology and immunohistochemistry? Only 4 animals in each group were used for the analyses.
3. Did the Authors perform negative controls in IHC procedures?
4. The significance level was set at p=0.05 (see chapter ''Statistical analysis''). Rather P<0.05 (Fig. 2, 10)?
Author Response
The manuscript presented for review is an interesting study on the molecular basis of recovery after ACLR. However, I have some questions and comments:
- There is no information in the chapter ‘’Biomechanical testing’’ about the type of test used in the research. In the description of the results there is information that it is a tensil test. There is also no name of the equipment used.
Author’s response: We have clarified that we carried out tensile testing to failure (line 133-134) using an Instron Materials Testing System (Norwood, Mass., USA) (line 128).
- How many slides and fields of view from each animal were evaluated for histology and immunohistochemistry? Only 4 animals in each group were used for the analyses.
Author’s response: It has been our standard to use 3-5 animals for histologic analysis in our 25+ years of working with animal models of connective tissue repair. Histologic analysis is generally semi-quantitative at best, and thus using 4 animals is rather typical.
We examined 8-10 fields of view for each specimen (line 144).
- Did the Authors perform negative controls in IHC procedures?
Author’s response: Yes, we routinely use negative controls where we omit the primary antibody and just use the secondary antibody and all subsequent reagents. This is done to check for non-specific binding of the secondary antibody or other reagents. This is routine in our laboratory, and is done to optimize the staining protocol for each antibody.
We have added the following to the revised manuscript: “Negative controls using just the secondary antibody and all subsequent reagents were used to verify the absence of non-specific staining” (line 154-155).
- The significance level was set at p=0.05 (see chapter ''Statistical analysis''). Rather P<0.05 (Fig. 2, 10)?
Author’s response: Thank you for noting this inconsistency. Yes, the p value was set at p<0.05. This has been changed (line 184).

Round 2
Reviewer 1 Report
The manuscript is now appropriate.